# Decline of Admission for Acute Coronary Syndromes and Acute Cardiovascular Conditions during COVID-19 Pandemic in Veneto Region

**DOI:** 10.3390/v14091925

**Published:** 2022-08-30

**Authors:** Marco Zuin, Giacomo Mugnai, Alberto Zamboni, Edlira Zakja, Roberto Valle, Giovanni Turiano, Sakis Themistoclakis, Daniele Scarpa, Salvatore Saccà, Loris Roncon, Francesca Rizzetto, Paola Purita, Angela Polo, Ivan Pantano, Antonio Mugnolo, Giulio Molon, Samuele Meneghin, Daniela Mancuso, Micaela Lia, Giuseppe Grassi, Ada Cutolo, Fabio Chirillo, Paolo Bozzini, Stefano Bonapace, Maurizio Anselmi, Gianluca Rigatelli, Claudio Bilato

**Affiliations:** 1Division of Cardiology, West Vicenza General Hospital, Via del Parco 1, 36071 Arzignano, Vicenza, Italy; 2Division of Cardiology, Legnago General Hospital, 37045 Legnago, Verona, Italy; 3Division of Cardiology, San Donà General Hospital, 30027 San Donà di Piave, Venezia, Italy; 4Division of Cardiology, Chioggia General Hospital, 30015 Chioggia, Venezia, Italy; 5Division of Cardiology, All’Angelo Hospital, 30174 Mestre, Venezia, Italy; 6Division of Cardiology, Santi Giovanni & Paolo Hospital, 30122 Venezia, Venezia, Italy; 7Division of Cardiology, Mirano General Hospital, 30035 Mirano, Venezia, Italy; 8Department of Cardiology, Rovigo General Hospital, 45100 Rovigo, Rovigo, Italy; 9Division of Cardiology, Verona University Hospital, 37100 Verona, Verona, Italy; 10Division of Cardiology, San Bassiano Hospital, 36061 Bassano, Vicenza, Italy; 11Division of Cardiology, IRCCS Sacro Cuore Don Calabria Hospital, 37024 Negrar, Verona, Italy; 12Division of Cardiology, Padua University Hospital, 35128 Padova, Padova, Italy; 13Division of Cardiology, Fracastoro Hospital, 37020 San Bonifacio, Verona, Italy; 14Division of Cardiology, Madre Teresa Hospital, 35043 Padova, Schiavonia, Italy

**Keywords:** acute coronary syndrome, COVID-19, epidemiology

## Abstract

Background: The present study aimed to examine longitudinal trends in hospitalizations for acute coronary syndrome (ACS) before and during the COVID-19 pandemic, by reviewing the data from 13 hospitals of the Veneto Region, in the north-east of Italy. Methods: We performed a multicenter, retrospective analysis including all the consecutive patients presenting with ACS and other acute cardiovascular (CV) conditions (defined as heart failure, arrhythmias, cardiac arrest and venous thromboembolism) hospitalized in 13 different hospitals of the Veneto Region covering a population of 2,554,818 inhabitants, during the first (between 15 March 2020 and 30 April 2020) and second (between 15 November 2020 and 30 December 2020) COVID-19 pandemic waves (the 2020 cohort). Data were compared with those obtained at the same time-windows of years 2018 and 2019 (the historical cohorts). Results: Compared to the historical cohorts, a significant decrease in the number of ACS cases was observed in 2020 (−27.3%, *p* = 0.01 and −32%, *p* < 0.001, comparing 2018 versus 2020 and 2019 and 2020, respectively). The proportion of patients hospitalized for acute CV conditions decreased during the first and second wave COVID-19 pandemic when compared to the historical cohorts (−36.5%, *p* < 0.001 and −40.6%, *p* < 0.001, comparing 2018 versus 2020 and 2019 and 2020, respectively). Pearson’s correlation evidenced a significant inverse relationship between the number of COVID-19 cases and both ACS hospital admissions (r = −0.881, *p* = 0.005) and hospitalizations for acute CV conditions (r = −0.738, *p* = 0.01), respectively. Conclusions: The decrease in hospitalizations for ACS and other acute CV conditions will strongly affect future patients’ management since undiagnosed nonfatal CV events represent a source of increased (and unknown) CV morbidity and mortality.

## 1. Introduction

During the first months of 2020, Italy became the first and most affected western country by SARS-CoV-2 infection. Throughout the first and second pandemic wave, due to the absence of vaccination and effective treatment for COVID-19 patients, the Italian government declared a general lockdown aiming to reduce the spread of the COVID-19 infection.

It is well known that urgent/emergent (i.e., unscheduled) hospitalizations greatly affect national and local health care resources and often cause transient overload of health care public system. On the other hand, scheduled/programmed admissions can be modulated according to current necessities and requirements of patients, populations and healthcare systems. During the COVID-19 pandemic, for example, hospital admissions were drastically reduced and/or postponed in Italy as in other countries, to: (a) preserve the hospital capacity for caring for COVID-19 patients and (b) reduce the risk of SARS-CoV-2 infection among patients and health care professionals. Surprisingly, the rate of acute coronary syndrome (ACS) also decreased in Europe and the US [1,2] in the same period, as did the rate of cardiovascular (CV) admissions for other acute reasons [3]. Therefore, we sought to examine longitudinal trends in hospitalizations for ACS and other cardiovascular conditions before and during the COVID-19 pandemic, by reviewing the data from 13 hospitals of the Veneto Region, in the north-east of Italy. Further, we investigated if a higher local burden of SARS-CoV-2 infection might inversely influence ACS and/or other CV condition hospitalizations.

## 2. Materials and Methods

### 2.1. Regional and Epidemiological Subset

The Veneto Region is located in north-east of Italy, with an extension of about 18.345 km^2^ and a population of over 4.9 million people. The first COVID-19 death was detected on 21 February 2020 in a 78-year-old man living in the municipality of Vò Euganeo, which is located near by the city of Padua, in the Veneto Region.

### 2.2. Study Population and Periods

This is a multicenter, retrospective study including all the consecutive patients presenting with an ACS or other acute CV condition (defined as heart failure, arrhythmias, cardiac arrest and venous thromboembolism) hospitalized in 13 different hospitals (Figure 1 and Appendix A) of the Veneto Region, covering a population of 2,554,818 inhabitants during the first (between 15 March 2020 and 30 April 2020) and second (between 15 November 2020 and 30 December 2020) COVID-19 pandemic waves (the 2020 cohort). Conversely, acute CV patients admitted over the same time-windows in 2019 and 2018 were considered as the historical cohorts. Diagnosis of ACS or other acute CV conditions were collected by the single Divisions of Cardiology, using patients’ medical records, clinical charts, hospital discharge letters or other medical documentation and compared with the related administrative codes according to the ICD-9-CM (International Classification of Diseases, nineth revision, Clinical Modification).

### 2.3. Statistical Analysis

Continuous variables were expressed as mean while categorical variables were presented as proportions and compared by the Pearson’s χ^2^ test. The Kolmogorov–Smirnov test was used to assess normality of continuous variables. Pearson’s correlation was carried out to evaluate a potential relationship between COVID-19 cases and ACS or acute CV hospitalizations. Statistical significance was defined as *p* < 0.05. Statistical analyses were performed using SPSS package version 20.0 (SPSS, Chicago, IL, USA).

## 3. Results

During the two 2020 pandemic waves, 843 ACS were hospitalized in the participating Divisions of Cardiology. In the same periods, 2195 patients were admitted because of other acute CV conditions. In the matching time windows of the years 2018 and 2019, the patients hospitalized with ACS or other acute CV conditions were 1160 and 3456 for the year 2018 and 1239 and 3964 for the year 2019.

Compared to the historical cohorts, a significant decrease in the number of ACS cases was observed in 2020 (from 1160 to 843, −27.3%, *p* = 0.01 and from 1239 to 843, −32%, *p* < 0.001, comparing 2018 versus 2020 and 2019 and 2020, respectively) (Figure 2, Panel A). Similarly, the proportion of patients hospitalized for acute CV conditions decreased during the first and second wave COVID-19 pandemic when compared to the historical cohorts (from 3456 to 2195, −36.5%, *p* < 0.001 and from 3694 to 2195, −40.6%, *p* < 0.001, comparing 2018 versus 2020 and 2019 and 2020, respectively) (Figure 2, Panel B).

Pearson’s correlation evidenced a significant inverse relationship between the number of COVID-19 cases and both ACS hospital admissions (r = −0.881, *p* = 0.005) (Figure 3, Panel A) and hospitalizations for acute CV conditions (r= −0.738, *p* = 0.01) (Figure 3, Panel B), respectively. This trend was mainly due to the drastic reduction in hospitalizations observed in the areas which suffered more severely by the COVID-19 spread, such as the areas around the cities of Venice, Verona and Padova.

## 4. Discussion

In 13 Divisions of Cardiology of the Veneto region, north-east of Italy, covering about two and half million inhabitants, a dramatic decrease both of ACS and other acute CV conditions was observed during the 2020 pandemic waves compared to the matched time periods of the previous years. These data were obtained directly by the clinical records of the single Divisions of Cardiology involved in the study and are in fully agreement with the data derived from the analysis of the administrative reports from the Veneto region [4]; overall, during the year 2020, there was a significant decrease in ACS with ST elevation compared to the 2018 and 2019 average, with a major decline (more than 25%) occurring during the first wave of the pandemic. Similarly, a significant decrease was observed regarding the hospitalizations for all the ACS especially during the first (about −29%) and the second (about −19%) wave.

The marked decline in hospitalizations for ACS and other acute cardiovascular conditions in the Veneto Region has been reported at national [1] and international level [2] and has been explained by the fact that people, fearing SARS-CoV-2 infection, might not seek medical attention even in case of severe symptoms. This observation seems to be supported by the higher number of out-of-hospital cardiac arrests observed during the COVID-19 pandemic in Italy [5]. Moreover, isolation at home during the lockdown along with the fear of being infected in the emergency department or during hospitalization may also have discouraged patients from seeking consultations with their general practitioner or their cardiologist [6].

On the other hand, it could be also hypothesized that there was reduced availability of the hospital facilities and the Divisions of Cardiology because of their conversion to COVID-19 hospitals and COVID-19 intensive care units. Indeed, during the COVID-19 pandemic, many hospitals and health facilities were converted to COVID-19 clinics all over the world. Most, if not all the 13 hospitals where the recruited Divisions of Cardiology were located, had ICUs or wards converted to COVID-19 facilities, as well. However, probably because the “intrinsic” nature of the cardiology wards/coronary care units (dedicated to patients with acute cardiovascular conditions), only one bed in a single Division of Cardiology was converted to a COVID-19 bed (Division of Cardiology of Rovigo Hospital). Our findings are in accordance with previous single-center and multicenter studies performed during the lockdown period in neighboring regions, demonstrating the same significant reduction in the rate of hospital admission for ACS and/or acute cardiovascular conditions [1,7,8]. However, our findings are derived from a larger population living in the first European region involved in the SARS-CoV-2 pandemic.

Our observation on the significant inverse correlation between the number of COVID-19 cases/severity of the spread of SARS-CoV-2 infection (from a geographical point of view) and hospital admissions for ACS and other acute cardiovascular conditions may support both the hypotheses. However, considering that only one bed of the whole ICU/ward beds of the 13 analyzed Divisions of Cardiology was converted and reserved for COVID-19 patients suggests that the fear of COVID-19 was the major reason for the decline in hospitalizations. The reduction in ACS hospitalization, however, may also be due to other reasons, such as the drop in air pollution observed during the lockdown period. Indeed, it is well recognized that one of the triggering factors of STEMI is the presence of air pollutants including sulfur dioxide, nitric dioxide, carbon monoxide, ozone, and particulate matter [9]. Notably, late or missed ACS treatment will imply future consequences not only in the single patient but also at the level of the entire national health system either in terms of health care costs or burden of cardiovascular complications. Probably, results will be different if we had considered the subsequent pandemic waves. Indeed, over time, the general population became less scared of COVID-19 due to lower mortality and diffusion of vaccines [10].

### Limitations

Our study has several limitations. Firstly, the retrospective design of the study, which is, however, partially mitigated by the large population analyses and the multicentric enrolment. Secondly, we did not include clinical endpoints such as in-hospital mortality or CV mortality, as this was not the aim of the study. Thirdly, the absence of clinical information regarding CV risk factors, ischemic time, previous CV events and treatments preclude any comparison between the patients enrolled during the different time-periods.

## 5. Conclusions

Whatever the reason, the observed decrease in hospitalizations for ACS and other acute cardiovascular conditions will strongly affect future patients’ management, since undiagnosed nonfatal CV events represent a source of increased (and unknown) cardiovascular morbidity and mortality.

## Figures and Tables

**Figure 1 viruses-14-01925-f001:**
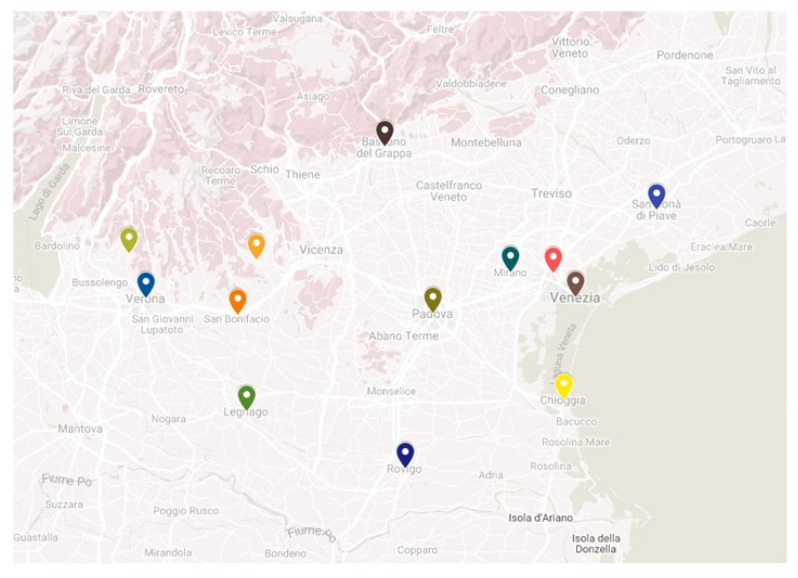
Location of enrolling centers.

**Figure 2 viruses-14-01925-f002:**
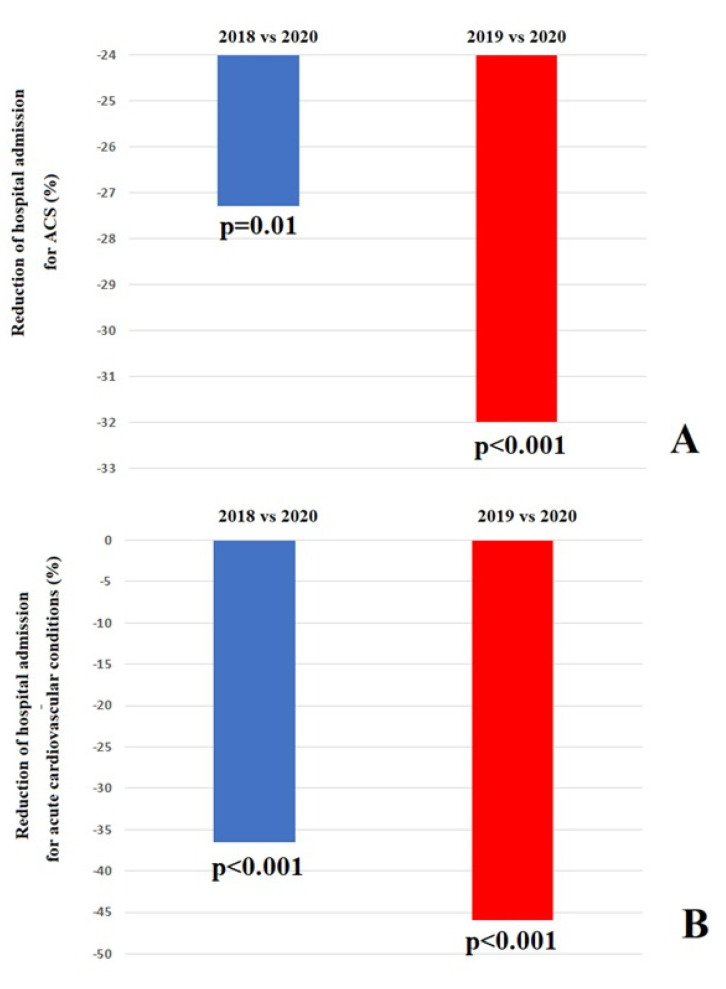
(**A**) Difference of hospital admission for acute coronary syndrome. (**B**) Difference of hospital admission for acute cardiovascular care.

**Figure 3 viruses-14-01925-f003:**
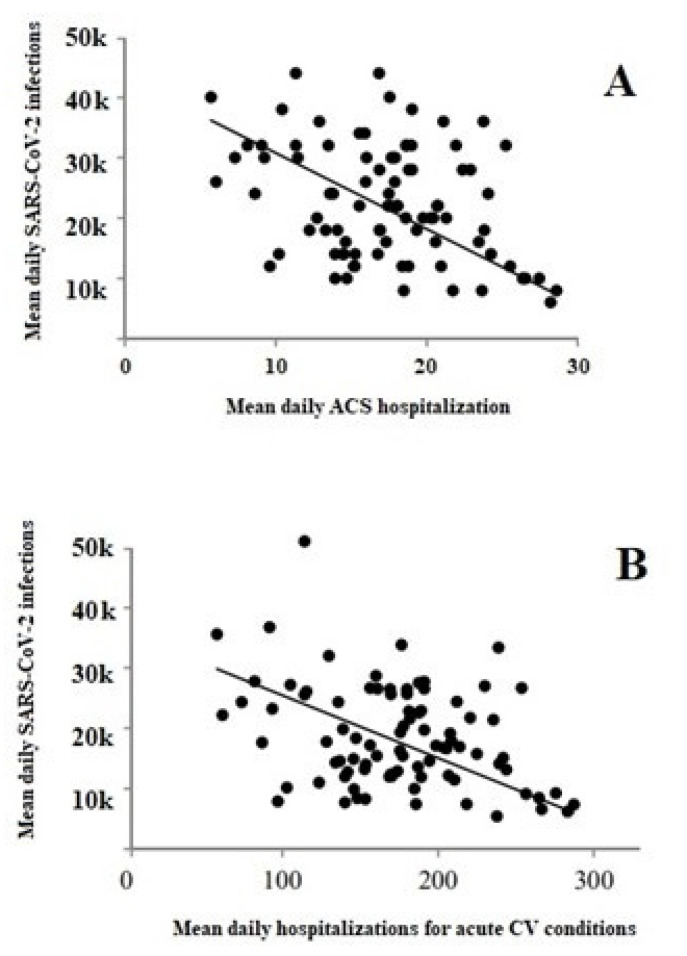
Pearson’s correlation between the number of COVID-19 cases and both ACS hospital admissions (Panel (**A**)) and hospitalizations for acute CV conditions (Panel (**B**)).

## Data Availability

Data are available, upon reasonable request, contacting the corresponding author.

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
