# Peer review of "Decline of Admission for Acute Coronary Syndromes and Acute Cardiovascular Conditions during COVID-19 Pandemic in Veneto Region"

_viruses, 2022, doi:10.3390/v14091925_

Round 1

Reviewer 1 Report

This paper shows an insight into the effects of the COVID 19 pandemic on cardiovascular diseases, in a particularly affected area in Italy. Even with the limitations of a reretrospective analysis, it shows important effects on hospitalisation for ACS and CV, which also gives hints for future organisational aspects in the pandemic era. Papaer is therefore worthy of publication in its current form. However, it should be specified what the authors mean by other acute CV diseases. 

Author Response

Answer: We thank the reviewer for the comments. We have defined the other acute CV both in the abstract and in the methods.

Reviewer 2 Report

Zuin et al performed a  multicenter, retrospective study on longitudinal trends in hospitalizations for acute coronary syndrome (ACS) before and during the COVID-19 pandemic in the Veneto Region.

Data on ACS cases observed in the first two COVID-19 pandemic waves of year 2020 were compared to an historical cohort of years 2018 and 2019. Authors found a significant decrease in the number of ACS cases in 2020 (-27.3%, and -32%, p<0.001, comparing 2018  vs 2020 and 2019 vs 2020, respectively.

1    1)  In the Discussion, authors made the hypothesis that the  fear of risking a close contact with infected COVID-19 pts might have discouraged the access to the emergency department in cases of chest pain.

I would suggest authors to consider also the fact that isolation at home due to the lockdown may have discouraged patients from seeking consultations with their general practitioner or their cardiologist.

2    2) Authors should also discuss the possible consequences of late or missed treatment in ACS patients in the next future for the entire health system.

Author Response

Zuin et al performed a  multicenter, retrospective study on longitudinal trends in hospitalizations for acute coronary syndrome (ACS) before and during the COVID-19 pandemic in the Veneto Region.

Data on ACS cases observed in the first two COVID-19 pandemic waves of year 2020 were compared to an historical cohort of years 2018 and 2019. Authors found a significant decrease in the number of ACS cases in 2020 (-27.3%, and -32%, p<0.001, comparing 2018  vs 2020 and 2019 vs 2020, respectively.

1)  In the Discussion, authors made the hypothesis that the  fear of risking a close contact with infected COVID-19 pts might have discouraged the access to the emergency department in cases of chest pain.

I would suggest authors to consider also the fact that isolation at home due to the lockdown may have discouraged patients from seeking consultations with their general practitioner or their cardiologist.

Answer: We thank the reviewer for the suggestion. We have added this aspect into the discussion.

2) Authors should also discuss the possible consequences of late or missed treatment in ACS patients in the next future for the entire health system.

Answer: We perfectly agree with the observation provided the reviewer. We have briefly discussed also the potential implications of missed ACS treatment for the entire national health care system.

Reviewer 3 Report

Datas from the proposed paper relatively to the decline in ACS hospitalizazione are well known. However the paper is interesting since it comes from one big region that was strongly hitten during the first wave but later than lumbardy. Paper can further improve with the following suggestion:

- In the discussion please comepare this data with similar one coming from lumbardy.

-  These results are mainly determined by movement limitation and fear of the population (indirect effets of the pandemia). Please better discuss this point also referring to the following review (10.1007/s40292-021-00464-8.)

- Results will probably be different id second wave will be included. In fact people were less scared and COVID-related mortality was lower (10.1007/s10389-021-01675-y.). Please discuss on this point in the relative section.

Author Response

Datas from the proposed paper relatively to the decline in ACS hospitalizazione are well known. However the paper is interesting since it comes from one big region that was strongly hitten during the first wave but later than lumbardy. Paper can further improve with the following suggestion:

- In the discussion please comepare this data with similar one coming from lumbardy.

Answer: We thank the reviewer for the suggestion. We have added a comparison with previous paper reporting the reduction of hospital admission for ACS performed in neighboring regions.

-  These results are mainly determined by movement limitation and fear of the population (indirect effets of the pandemia). Please better discuss this point also referring to the following review (10.1007/s40292-021-00464-8.)

Answer: We perfectly agree with the reviewer. We have added this issue into the discussion and cited the mentioned paper.

- Results will probably be different id second wave will be included. In fact people were less scared and COVID-related mortality was lower (10.1007/s10389-021-01675-y.). Please discuss on this point in the relative section.

Answer: We perfectly agree. We have discussed this issue as suggested.

Round 2

Reviewer 3 Report

Authors replies to all the query raised and paper improves and can now be accepted for publication.